# Development of a Functional Acceptable Diabetic and Plant-Based Snack Bar Using Mushroom (*Coprinus comatus*) Powder

**DOI:** 10.3390/foods12142702

**Published:** 2023-07-14

**Authors:** Maria Dimopoulou, Patroklos Vareltzis, Stylianos Floros, Odysseas Androutsos, Alexandra Bargiota, Olga Gortzi

**Affiliations:** 1Department of Agriculture Crop Production and Rural Environment, School of Agricultural Sciences, University of Thessaly, 38446 Volos, Greece; mdimopoulou@uth.gr; 2Laboratory of Food and Agricultural Industries Technologies, Chemical Engineering Department, Aristotle University of Thessaloniki, 54124 Thessaloniki, Greece; pkvareltzis@cheng.auth.gr (P.V.); florstyl@cheng.auth.gr (S.F.); 3Lab of Clinical Nutrition and Dietetics (CND-Lab), Department of Nutrition & Dietetics, School of Physical Education, Sport Science and Dietetics, University of Thessaly, 42132 Trikala, Greece; oandroutsos@uth.gr; 4Department of Endocrinology and Metabolic Diseases, Faculty of Medicine, School of Health Sciences, University Hospital of Larissa, University of Thessaly, 41334 Larissa, Greece

**Keywords:** functional foods, diabetics, snack bars, sensory analysis, chemical analysis, in vitro digestion

## Abstract

Consumers’ growing concern about health and well-being has led to increased interest in functional foods. This research aims to evaluate the physicochemical and antioxidant properties of a functional plant-based (PB) snack bar enriched with *Coprinus comatus* powder. The snack bar formulations exhibited a wide range of flavor and textural characteristics. Two PB snack bars and four commercial bars were evaluated by a consumer panel of healthy volunteers (*n* = 20). The PB snack bar scored ‘like extremely’ on the 9-point hedonic scale. External preference mapping determined that sweetness, flavors, cohesiveness, gumminess, and adhesion had the greatest influence on consumer acceptability. Water content, ash, protein, fat, carbohydrate, reducing sugar, resistant starch, and dietary fiber were measured. Nutritional content was enhanced (omega 3, fiber and protein), and samples were shelf life stable (aw < 0.29; moisture content < 10%). In addition, the PB snack bar underwent simulated digestion according to the INFOGEST protocol, and from the comparative evaluation, the PB snack can be seen to control the post-prandial glycemic responses, as observed by the different degree of reducing sugars released via the matrix. The PB snack bar can be further functionally enhanced by the addition of their unique ingredients such as *Coprinus comatus*. *Coprinus comatus* powder is claimed to benefit glycemic control in diabetes and has attracted growing interest in terms of its potential use in natural products with possible health benefits.

## 1. Introduction

The number of people with Diabetes Mellitus (DM) is increasing dramatically worldwide in both developed and developing countries and is expected to reach 700 million by 2040 [1]. An alarming trend of increasing type 2 DM, which is a consequence of aging populations and the continuous increase in obesity, should be noted [2,3].

The complications of DM can be categorized into two types: acute and chronic, also known as long-term complications. Acute complications include hypoglycemia and diabetic coma, which can be life threatening [4]. Long-term complications include increased cardiovascular disease risk, peripheral vasculopathy and neuropathy, retinopathy, blindness, and kidney disease [5,6].

Chronic hypoglycemia is thought to be a major factor that leads to these complications, and it occurrs through several intracellular mechanisms, including overactivation of the polyol pathway, activation of protein kinase C, increased hexosamine pathway, and increased formation of advanced glycation end products (advanced glycoxidation end products—AGE).

All of these mechanisms are considered to lead to tissue damage through a common pathway, i.e., increased oxidative stress (Oxidation stress—OS) [7]. Increasingly, data demonstrate the link between the intrinsic regulation of oxidative metabolic pathways and the onset or progression of either DM or its complications [8]. Among these processes, lifelong cumulative glycosylation or the formation of advanced glycation end products (AGEs) contribute significantly to OS, redox-sensitive cellular metabolic transcription factor overactivity, and, ultimately, inflammatory reaction or damage. However, neither the magnitude nor the origin of these increased levels—glycosylation end products—that favor oxidations are fully understood [9].

Human studies have recently demonstrated significant associations between dietary AGEs, circulating AGEs, and several markers of inflammation [10]. Reviewing the AGE content in common foods may prove to be a feasible and broadly applicable intervention in both diabetic and healthy individuals.

The international literature proves that healthy dietary interventions can be as effective as pharmaceutical interventions in terms of improving health indicators (e.g., obesity, glucose, glycated hemoglobin, lipid profile, etc.) in diabetic patients, and they improve indicators without carrying the risks of health implications related to the side effects of drugs [11].

Recently, the study of molecular mechanisms of biofunctional food ingredients provided evidence that they have beneficial effects on carbohydrate metabolism, β-cell function and insulin resistance, lipid and lipoprotein metabolism and adipose tissue function, oxidative balance, inflammatory response, body weight management, micro- and macro-vascular diseases, and induction of gene expression [3]. Meal planning based on these healthy foods can be used as an effective strategy to manage multiple health parameters in patients with diabetes and cardiovascular disease [12]. Bioactive ingredients that simultaneously lower glucose and lipid levels include soluble fiber (mainly β-glucans and pectin), polyphenols, and alkaloid berberine. In addition, according to the American Diabetes Association, a diet rich in omega-3 fatty acids can lower triglyceride levels in diabetics with hyper-triglyceridemia. In particular, in regard to olive oil, its bioactive components, such as monounsaturated fatty acids, and its main polyphenols, such as hydroxytyrosol and oleoeuropein, have been associated with the prevention of inflammation and oxidative stress, reduction in glucose levels and carbohydrate absorption, and increase in insulin sensitivity and associated gene expression [6]. Also, whole-grain products have been shown to be effective in improving glycemic control and lipid profile in patients with type 2 diabetes [13].

According to the recent recommendations of the European Association for the Study of Diabetes (EASD), nutritionally complete low-energy formula products can be used by patients either for temporary weight-loss induction as ‘total diet replacement’ (replacing all meals) or by replacing 1–2 meals/day. Replacing 1 meal/day or 3–6 meals/week can also be used for maintenance of longer-term weight-loss [14]. Additionally, there is much public and scientific interest in whether eating foods containing on-sugar sweeteners (NSS)—which contain few or no calories—should be recommended as a strategy to reduce consumption of free sugars [15]. Recently, the WHO warned that the consumption of free sugars has been linked to escalating rates of overweightness and obesity, as well as development of diet-related non-communicable diseases (NCDs), including dental caries, type 2 diabetes, cardiovascular diseases, and cancer [16]. It was very important that the new PB snack bars used a natural product as a sweetener: the carbon honey. Moreover, the mushroom *Coprinus comatus* is claimed to benefit glycemic control in diabetes [17] and attracts growing interest in terms of its potential use in natural products with possible health benefits.

To date, very few studies [18,19] have examined whether the utilization of new bio-functional foods can lead to glycemic control and the improvement of other biomarkers (e.g., lipid profile) in diabetics. In previous studies, biofunctional ingredients included fiber, polyphenols, and alkaloid, but not proteins. Therefore, rice, pea protein and *Coprinus comatus* powder were selected as biofunctional ingredients. Based on these protein sources, two plant-based snack bars were developed: one snack bar used rice-protein and one snack bar used pea protein; in both bars, the protein content was further increased by adding the powder made of the mushroom *Coprinus comatus*.

## 2. Materials and Methods

### 2.1. Materials

All ingredients, as well as the commercial snack bars used in the study, were purchased from a retail supermarket, except for *Coprinus comatus* powder, which was obtained from the Museum of Mushrooms, Meteora, Greece.

### 2.2. Preliminary Recipe Development

A thorough bibliographic review was carried out to gather information on the formulation of products that are already commercially available and claim to help in controlling glucose levels (Table 1). The last novel product [20] was the starting point for the preparation of our own innovative products, modifying the composition of the two snack-bars to almost equal the nutritional value of the bars that have already been studied in terms of nutrient ratios.

Guidelines for the method of preparation were obtained from a typical snack bar formulation [24], and the dry and wet ingredients were then mixed, molded, and baked at 130°/15 min. Filling ingredients were heated with stirring to create 84–86% soluble solid content and were placed between base parts. The PB snack bar formulations were previously tested for complete agglomeration of solid ingredients, and the best of them are shown in Table 2. The new plant-based bars were developed in the Laboratory of Technology and Quality Control and Food Safety of the Department of Agriculture, Plant Production and Rural Environment of the University of Thessaly according to the rules of Good Hygienic Practice (GHP). The bars used for the sensory evaluation were produced on a pilot scale according to ISO 22000. For the preliminary experiments, ten bar prototypes were formulated and consisted of 44% (*w*/*w*) oat flakes and oat bran (ratio 80:20); 2% natural sweeteners and flavors (agave syrup, carob honey, and fruits—not only dried, but also juice, zest and cinnamon); sources of fat, i.e., almonds, flaxseed, and sunflower oil, were approximately 20%, 1% and 14%, respectively; and 19% plant-based protein (≈18% either rice or pea protein and a small percentage of mushroom protein). The bar’s ingredients were as follows: pea or rice protein [25], agave syrup [26], carob honey [27], oat bran, oat flakes [28], lemon juice and zest [29], orange juice and zest [30], Coprinus comatus powder [31], almonds [32], flaxseed [33], sunflower oil [34], cranberries [35,36], apples [37], cinnamon [38], and a chocolate coating [39] with stevia [40]. All of the above ingredients help to lower and control blood sugar. The macronutrient profile of the plant-based snack-bars was analyzed according to methods described by the Association of Official Analytical Chemists International (2021) [41], and samples were shelf life stable (aw < 0.29; moisture content < 10%) [42]. Kjedahl’s method of protein determination was used for cereal bars, the Titrametric Lane–Eynon method was used for determination of sugars, GC-FID was used determination for saturated and unsaturated fat, and, finally, an enzyme-linked immunosorbent assay was used for determination of total aflatoxins (<2 μg/Kg). Functionality could be attributed to the fact that the ingredients have high protein and dietary fiber content according to EFSA [43].

Protein content was found to range between 4.3 g/100 g and 19.5 g/100 g, carbohydrate content ranged between 38.4 g/100 g and 60.6 g/100 g, and fat content ranged between 6.8 g/100 g and 22.2 g/100 g. The nutritional values of different commercial snack-bars are given below (Table 3).

The list of all snack bars for analysis:Snack-bar 1: rice and *Coprinus comatus* powder protein plant-based snack-bar;Snack-bar 2: pea and *Coprinus comatus* powder protein plant-based snack-bar;Snack-bar 3: commercial diabetic snack bar;Snack-bar 4: commercial snack-bar with reduced sugar and high fiber content, as well as cranberries (Control sample/Euphoria meals);Snack-bar 5: commercial snack-bar with reduced sugar and high fiber content, as well as coconut chocolate (Euphoria meals);Snack-bar 6: commercial snack bar with reduced sugar and high fiber content, as well as peanut butter (Euphoria meals).

All snack bars selected for analysis included a statement or claim that promoted the healthy or nutritional value of the product, such as snacks with a reduced amount of the following unhealthy ingredients: reduced sugar, less fat, no preservatives, or additives; snacks with the following ingredients that promote health: omega-3, wholegrain, plant protein, and fibers; and snacks developed for the following specific objective or diet: diabetes. Diabetic food for high blood glucose or diabetes mellitus consists of edible products rich in refined carbohydrates and low sugar content [22,23,44,45,46]. Driving factors are as follows:Growing consumer awareness;Rise in preventive measures taken by consumers;Increasing use of artificial sweeteners.

### 2.3. Identification of Attributes Important for Snack Bar Acceptability

The organoleptic characteristics of the snack bars were assessed subjectively by a sensory panel using organoleptic acceptance questionnaires and objectively via texture profile analysis (TPA analysis). Furthermore, the snack bars underwent simulated digestion based on the INFOGEST protocol to assess its oxidative stability [47].

#### 2.3.1. Texture Profile Analysis

Texture profile analysis (TPA) has been widely used in food-product characterisation and quality control since it was invented by the General Foods Corporation’s Technical Centre, and it was conducted with a Brookfield CT3-4500 Texture Analyzer [48]. The operational parameters for the TPA test were as follows: probe—TA41 (6 mm diameter cylinder probe); tigger value—4.5 g; deformation—5 mm; and Speed—1.2 mm/s. Parameters recorded in the texture profile analysis, such as hardness, adhesiveness, and cohesiveness, were widely used for comparison with the sensory attributes [49].

#### 2.3.2. Sensory Analysis

The literature on sensory methods of texture evaluation contains fragmented information regarding definitions of texture and panel techniques. In our study, the panel consisted of 20 people, who were made up of 14 men and 6 women aged between 20 and 65 years old. The separation of ages was performed according to their age decade, namely 20–30 years, 31–40 years, 41–50 years, 51–60 years and >61 years. To calculate the subjective perception and personal desire, the organoleptic acceptance, and the nutritional value of the snack bars, a questionnaire was used (Adapted from UTT, BAFT, B.Sc. Food Science and Technology, Student Project for PROJ2005 Capstone, 2012) [50]. Consumers (*n* = 20) visited the Department of Nutrition and Dietetics, School of Physical Education, Sport Science and Dietetics, and occasionally ate snack bars. Focus groups were led by an experienced facilitator and lasted 60 min, and participants responded to ‘dislike’ or ‘like extremely’ on the 9-point hedonic scale regarding their implied acceptance of six snack bars, two of which were prepared in the laboratory and four of which were commercial samples (Table 3). Specifically, they were asked to rate the samples on the basis of a 9-point hedonic scale anchored by the following scores: 1 = ‘Disliked extremely’; 2 = ‘Disliked very much’; 3 = ‘Moderately disliked’; 4 = ‘Slightly disliked’; 5 = ‘Indifferent’; 6 = ‘Slightly liked’; 7 = ‘Moderately liked’; 8 = ’Liked very much’; and 9 = ‘Liked extremely’. Each sample snack bar was wrapped in a resealable plastic snack bag and labeled with a random 3-digit number. Samples were presented one at a time. Purified water was available to cleanse the palate as required.

### 2.4. Static In Vitro Simulation of Gastrointestinal (GI) Digestion

#### 2.4.1. Enzyme Activity Assays

The oxidative behavior of samples along the GI tract was studied by implementing the INFOGEST protocol. The INFOGEST protocol was divided into 3 stages, starting with the preparation of the samples, which included the characterization of the activities of enzymes and bile salts used. The activities of all enzymes purchased are given, except for pancreatin, which was calculated according to the pancreatin assay of the INFOGEST protocol. The characterization of pancreatin activity was normalized to the trypsin and pancreatic lipase activity assays [47].

#### 2.4.2. Stock Solution Preparation

According to the INFOGEST protocol, digestion involved the exposure of food to three phases (oral, gastric, and intestinal phases). The electrolytes used for each stage were prepared in advance in stock solutions and stored at −10 °C. Specifically, stock solutions of KCl (0.5 M), KH_2_PO_4_ (0.5 M), NaHCO_3_ (1 M), NaCl (2 M), MgCl_2_(H_2_O)_6_ (0.15 M), (NH_4_)_2_CO_3_ (0.5 M), HCl (0.09 M), and CaCl_2_(H_2_O)_2_ (0.025 M) were prepared. These stock solutions were used to create simulated fluids (1.25×) for each stage of digestion, known as simulated salivary fluid (SSF), simulated gastric fluid (SGF), and simulated intestinal fluid (SIF), as described in the INFOGEST protocol manuscript [47].

#### 2.4.3. Oral Digestion Phase

The protocol began with the preparation of the samples and their primary homogenization. Firstly, 1 g of food sample was added to a test tube and mixed with SSF (1.25×). Distilled water was added to achieve a final volume ratio of 1:1. The final mixture was transferred to a heated incubator, where the test tube was shaken under heating for 2 min at a constant temperature of 37 °C [47].

#### 2.4.4. Gastric Digestion Phase

The oral bolus was mixed with SGF (1.25×). Additionally, pepsin was solubilized with water to reach a final activity of 2000 U/mL and added to the mixture. The pH was set at 3 via the addition of the HCl solution (1 M). Distilled water was added until a final volume ratio of 1:1 was reached. The final mixture was transferred to the temperature-controlled incubator, where it remained for 2 h at a temperature of 37 °C [47].

#### 2.4.5. Intestinal Digestion Phase

The gastric chyme was mixed with SIF (1.25×), pancreatin solution (100 TAME U/mL), and bile salt solution (10 mmol/L). The pH was set at 7 via the addition of NaOH solution (1 M). Distilled water was added until a final volume ratio of 1:1 was reached. The mixture was, finally, transferred to the temperature-controlled incubator, where it remained for 2 h at a constant temperature of 37 °C [47]. When needed, BHT (500 ppm) was added after the end of the protocol to inhibit further oxidation, while samples were frozen and stored at −20 °C until further evaluation.

### 2.5. Determination of Lipid Oxidation

#### 2.5.1. Peroxide Value (PV)

The PV was measured according to a modified method, as described by Richards et al. [51]. Firstly, 1 g of the lipid sample was mixed with 10 mL of CHCl_3_-CH_3_OH (2:1, *v*/*v*). Next, 500 ppm of BHT was added to the test tube to stop the oxidation process. The mixture was homogenized for 15 s and filtered to remove solids. In total, 1.5 mL of NaCl (0.5%) was then added, and the mixture was vortexed and centrifuged at 4000 rpm for 10 min to separate the two phases at ambient temperature. After centrifugation, the lower phase of CHCl_3_ was collected, and a quantity of CHCl_3_-CH_3_OH (2:1) was added until a final volume of 10 mL was reached. Lastly, 25 μL of NH_4_SCN solution (30% *w*/*v*) and 25 μL of FeCL_3_ solution (0.66% *w*/*v*) were added, and the mixture was vortexed for 2–4 s. Peroxides were measured via spectroscopic absorption at 500 nm using a Quartz cell. As a blind sample, 10 mL of 2:1 CHCl3-CH3OH mixture was used; the oxidation products were expressed in meqO_2_/kg of lipid phase using a standard curve formed using cumene hydroperoxide solutions [52,53].

#### 2.5.2. Thiobarbituric Acid (TBARS) Method

The TBARS were determined according to Lemon with small modifications. In brief, an initial amount of 1.5 g of the sample was added to a test tube containing 5 mL of TCA (7.5% *w*/*v*). The mixture was homogenized, vortexed, and centrifuged for 25 min at 4000 rpm. An aliquot of 2 mL was mixed with 2 mL of TBA solution (0.02 M). The mixture was heated in a water bath for 40 min at a constant temperature of 100 °C. Finally, the absorbance was measured spectroscopically at 532 nm. As a blind sample, TBA:TCA solution (1:1) was used, and the oxidation products were expressed as MDAeq (μmol/L) using a standard curve constructed using TEP solutions [54].

### 2.6. Statistical Analysis

Statistical analysis was performed using the Statistical Package for the Social Sciences (SPSS 21). A frequency analysis was performed for each of the variables in the questionnaires [50]. Descriptive statistics of continuous variables were expressed as mean ± standard deviation and percentages for each category. In addition, the t-test for independent variables was used to investigate any differences between the two sexes (male–female) of the research participants. The level of statistical significance was set at *p* < 0.050. A one-way ANOVA was carried out using R software (v. 3.6.3, R Core Team, Vienna, Austria) to determine significant differences between the stages of in vitro gastrointestinal digestion and the characteristics of the Texture profile analysis; the significance was accepted at a *p*-value of < 0.05. Finally, Fisher’s LSD pairwise comparison was performed on the data.

## 3. Results

### 3.1. Selection and Quality of Six Bar Samples

#### Nutritional Value

Besides vitamins and minerals, snack bars are also a good source of proteins and carbohydrates (Figure 1). Their protein content varies between 4.3 and 19.5%, as shown in Table 3. Mushroom proteins contain most essential amino acids. Protein bars are also rich in carbohydrates. The total carbohydrate content in the snack bars varies from 60.6% (commercial snack bar with reduced sugar and high fiber content snack bar, as well as coconut chocolate) to 38.4% (plant-based snack bar with pea protein). PB snack bars are mainly composed of mannitol, glycogen, and hemicellulose, as well as a smaller amount of reducing sugars. Mushrooms are rich in various vitamins and minerals that many vegetable and meat products lack or contain only in low concentrations, such as Vitamin D and selenium. Even though baking is one of the best ways of extending the shelf life of snack bars, vitamins can be lost in the process. Environmentally friendly packaging improves the preservation of total sugars, ascorbic acid, and bioactive compounds during storage by reducing moisture loss. However, there is limited information on the effects of chemical processes, package ageing, pulsed electric field, and ultrasound on the nutrient composition and bioactive properties of mushrooms [55]. Research has shown that the nutrient composition of different fungal species varies slightly. Protein content ranges between a minimum of 4.3 g/100 g (commercial diabetic snack bar) and a maximum of 19.5 g/100 g (pea-protein plant-based snack bar). Carbohydrate content ranges between a minimum of 38.4 g/100 g (Pea-protein plant-based snack-bar) and a maximum of 74 g/100 g (commercial diabetic snack-bar). Fat content ranges between a minimum of 6.8 g/100 g (commercial diabetic snack bar) and a maximum of 22.2 g/100 g (pea-protein plant-based snack bar). Plant-based snack bars may be a great “anti-diabetic” group of plant foods, as they contain complex carbohydrates, which the body slowly metabolizes, causing blood glucose levels to rise gradually. Common characteristics of the above bars are their increased fiber content and, as a consequence, their low glycemic index [55].

### 3.2. Acceptability, Purchase Intent and ‘Just Right’ Responses by Consumer Panel

Descriptive analysis of the survey results showed that 65% of participants (Question 3) consume plant-based foods. However, the frequency of consumption (Question 4) varies, with 40% of participants reporting frequent consumption of such products. Most participants (65%) indicated that they are somewhat satisfied with the bars offered by the market (Question 5). This result also underlines the importance of this research, as it focuses on the satisfaction associated with consuming a granola bar. For most respondents (Question 7), it is not easy to find granola bars on the market (40% and 45%) that have the characteristics they desire. For the next two questions (Question 8 and Question 9), respondents stated that they would consistently (100%) buy bars with the desired characteristics on the market if they could find them, regardless of the brand or company that offered them.

The questions listed in the second part of the questionnaire assessed the organoleptic acceptability and nutritional value of the texture of the granola bar. In addition to the frequency analysis of the perceived rating by all participants of the granola bars, a one-way ANOVA statistical analysis was used to determine the differences between the two genders of participants (male and female). For organoleptic acceptability of the rice protein bar, statistically significant differences were found between males and females for grittiness (*p* = 0.34), elasticity (*p* = 0.035), sticky taste (*p* = 0.039), chewy texture (*p* = 0.034), moistness (*p* = 0.047), and overall texture characterization (*p* = 0.021) in favor of males. No statistically significant differences were found between males and females for the other ratings. The graph below (Figure 2) shows the total degree of acceptance, considering all 4 organoleptic characteristics based on the liking questionnaires (appearance, aroma, texture, and taste).

### 3.3. Sensory Attribute Intensities via Texture Profile Analysis

Texture profile analysis (TPA) is an instrumental test that provides objective measurements of texture parameters, which are major factors in food acceptability.

Τhe newly developed snack bars (1 and 2) exhibited high organoleptic acceptance scores of 9 and 8, respectively (Figure 2). These two bars also exhibited significantly higher values of hardness and chewiness (Table 4) according to the TPA analysis, indicating that these two attributes significantly contributed to the score they were allotted by the panelists. One-way analysis of variance was used to examine whether there were statistically significant differences between the snack bars’ characteristics. The results show that there were statistically significant differences between the snack bars regarding the following textural characteristics: hardness F(3,15) = 194.7, *p* < 0.000; springiness F(3,15) = 9.539, *p* < 0.002; adhesiveness F(3,15) = 7.210, *p* < 0.005; chewiness F(3,15) = 24.803, *p* < 0.000; and gumminess F(3,15) = 40.369, *p* < 0.000. Another important quality characteristic was the cohesiveness of the samples, as the new bars exhibited similar or even better values than the other bars. On the other hand, the peanut butter snack bar scored the lowest values in the texture attribute, according to the panelists. This result was accompanied by the lowest value in chewiness and springiness, according to the TPA analysis. Previous research into snack bars showed a significant positive correlation between instrumental and sensory texture analysis for hardness, springiness, and adhesiveness [56,57]. Snack bar 1 has significantly higher chewiness, hardness, and gumminess characteristics. The ingredients that cause these observations are plant-based proteins, which have adequate functional properties, such as an emulsifying ability, fat-absorbing capacity, gelling, and water-holding ability [58].

### 3.4. Comparative Evaluation Regarding Bioavailability (In Vitro Digestion) and Nutritional Value

In all cases, there is a hydrolysis of the proteins as the stages of digestion progress (to a slightly different degree depending on the bar), resulting in a decrease in the percentage of protein (the Lowry method is based on the reaction of peptide bonds; a greater degree of hydrolysis essentially means fewer peptides ties). The bar with cranberries shows more stable behavior at the lowest degree of hydrolysis. Hydrolysis can be interpreted either positively (e.g., smaller protein size and, thus, more easily digestible) or negatively (loss of original structure and, therefore, functionality).

One-way analysis of variance was used to examine whether there were statistically significant differences between the snack bars during simulated digestion in three phases. The results show that there were only statistically significant differences between the snack bars based on the protein content before oral digestion (t = 0 min), after gastric digestion (135 min), and at the intestinal phase (195 min), while for sugars, there were significant differences among the six stages during simulated digestion.

It is known that the harsh conditions during digestion accelerate lipid oxidation in foods. The oxidation rate was followed by two methods that determined both peroxide and secondary oxidation products. In most cases, PVs were formed in the gastric phase and TBARS were formed in the intestinal phase. Primary oxidation was stunted for the plant-based snack bars, but oxidation began after 135 min when lipases acted to further oxidize fats. The bars had a higher content of unsaturated fats, which are known to be more oxidizable, though in the developed bars, they were better protected due to the presence of other antioxidant components, and their oxidation occurred in the time range of 190 to 245 min. On the other hand, the PVs for rice bars were almost flat and stable for the duration of digestion, while the commercial diabetic bars showed a clear peak at about 75 min.

Also, reducing sugars were determined at two different time intervals during the gastric and intestinal phases of digestion. The most important result of the above analyses is the generation of values of total reducing sugars in the intestine. The values of reducing sugars per gram/bar (in the intestine) in the developed products (rice protein bar and pea protein bar) compared to commercial bars (diabetic bar and bar with cranberries or coconut chocolate or peanut butter) were 0.307, 0.204, 0.48, 0.358, 0.45, and 0.51 g, respectively. From the comparative evaluation, the numbers of reducing sugars in the stomach and intestine are lower in the developed bars than in the other commercial bars.

An important finding is that although all bars have approximately the same level of reducing sugars at the beginning, the commercial diabetic bar shows a distinct peak in the stomach at about 75 min, and a different level of sugar release from the matrix is observed in the new products. Bars with high protein content appear to be better at protecting carbohydrates from hydrolysis, according to the nutritional profile, and the concentration of reagents remains low. In particular, the rice bar shows the best behavior in terms of sugar reduction (Table 5), and it is probably more suitable for diabetes. The traditional approach to diabetes management focuses on limiting refined sugars and foods that release sugars during digestion—that is, starches promote overall dietary wellness and glycemic control and prevent or ameliorate diabetes-related complications [59].

## 4. Discussion

Patients with diabetes are advised to follow specific dietary recommendations [60], which may lead to beneficial changes in biochemical indicators, such as fasting glucose, blood glucose, and overall health. The consumption of functional foods may support them in adhering to these recommendations [59,61]. In this sense, ready-to-eat nutritious products, such as snack bars, are highly appreciated for their convenience [24]. Snack products that use plant-based wholesome ingredients have the potential to improve health effects, including glycaemia, satiety responses, and lipid metabolism [23]. Firstly, the goal of food design for the PB bars was to find healthy ingredients and include them in proper portions. With in-depth research into plant-based protein, *C. comatus* is a mushroom with rich nutritional value [62] that provides innovative therapeutic benefits for diabetic patients, [31] as well as antioxidant effects [63]. It is being used for the first time in the functional food market. Overall, the results indicate that replacing sources of animal protein with plant protein leads to modest improvements in glycemic control in individuals with diabetes [64]. Plant-based protein benefits diabetic people as it is rich in fiber and low in saturated fats. This characteristic of plant-based proteins helps to reduce the body’s resistance to insulin [65]. Rice and Pea proteins are promising substitutes because of their “allergen-friendly” nature, as well as their emergence in the food market. However, manufacturers generally provide limited functionality information about these proteins [66].

The effect of protein on glucose and insulin responses depends on its digestibility [67,68]. In general, both digestion and absorption of nutrients in the small intestine are rapid processes, and in vitro starch and protein hydrolysis (Table 5 and Table 6) during all of the phases of simulated digestion revealed better absorption of the newly developed products, especially rice- and *Coprinus comatus*-protein snack bars. Moreover, in a recent study, the deconvolution of the amide I band of the Raman spectra indicated that pea proteins were different from rice proteins, which may be the reason for differences in our results, along with the different functionalities of the two PB proteins [66]. Except for their protein content, cereal bars [18,22,23,46,69,70] can be a great “anti-diabetic” group of plant foods due to their increased fiber content and low glycemic index [55]. A review emphasized that the regular consumption of snack bars enriched with nuts reduced average fasting blood sugar levels by up to 11% and decreased post-meal blood sugar levels by up to 14% in patients with diabetes [21].

One more interesting feature of the developed PB bars is their high content of antioxidants, sourced both from the fruit and the carob honey. Mushroom protein can also play a protective role against lipid oxidation. The possible role of these ingredients against lipid oxidation was demonstrated by measuring the PVs and TBARS. A diet rich in antioxidants limits age-related weight gain and insulin resistance [10]. The antioxidant profiles of the snack-bars were attributed to the greater amount of polyphenols sourced from the berries [70], though in our study, many ingredients played a role, such as the mushroom powder, the cranberries, and the fruit juices and flavors (cinnamon). The previous study [70] determined whether polyphenol-rich fruits added to carbohydrate-based foods produce a dose-dependent moderation of oxidative stress responses, and the results were similar in our study. A significant finding was made regarding the plant-based bars containing *Coprinus comatus* powder, which exhibited superior peroxide values than both commercial bars and those marketed as “healthy” or “diabetic snacks.”. According to the results (Table 6) from the in vitro digestion 75 min after the consumption of the three mentioned diabetic bars, the difference was extreme and emphasized the superiority of the developed products.

Many kinds of snacks have high Glucose Index (GI) scores and can cause a spike in blood sugar levels. Thus, for patients with diabetes, many developed products either prevent hypoglycemia or reduce post-prandial hyperglycemia [21,71]. Nite Bite Timed Release Glucose Bar (ICN Pharmaceuticals, Inc.), Ensure Glucerna [21], and Nothing Else- Snack (Re)formulation (ΝΕ) [20] are only a few of the diabetic snack bars available in the market. The nutrition profile for carbohydrates ranged, on average, from 15–44.9 g, while in the case of the new product, it was estimated at approximately at 39 g and had a higher protein percentage than other products. Expected health benefits, such as regulation of glucose metabolism, regulation of satiety, and reduction in low-density lipoprotein, were mainly related to plant-based protein content and high antioxidant content [72,73]. Additionally, these new products are necessary not only due to the lack of similar diabetic products, but also because of the lower cost of producing the plant-based bars [74].

According to the International Diabetes Federation [75] 5–20% of total health costs are spent on diabetic health complications. The Mediterranean Diet, low-carbohydrate diets, and plant-based diets are all examples of nutritious eating patterns that have demonstrated metabolic advantages [76,77]. As a therapeutic target, it is also important to maintain a normal weight by ensuring the intake of all macro- and micro-nutrient compounds [78]. The development of foods like our plant-based snack bar, which is of high nutritional value (high in fiber, protein, omega-3 fatty acids, and natural antioxidants), is very important, according to the dietary recommendations of the American Diabetes Association [13], in order to prevent and slow the complications of type 2 DM [13]. According to the Hellenic Diabetes Association, a moderate intake of simple sugars (up to 50 g/day) can be included in the diet of people with DM1 and DM2, while the total percentage of simple sugars should not exceed 10% of the 55% of daily carbohydrate intake. Finally, a moderate intake of fructose (up to 30 g/day) does not appear to have adverse effects on insulin and plasma lipids in T2DM subjects [79]. It has also been found that many people with DM have low levels of antioxidant intake. Supplemental administration of antioxidants through biofunctional foods is important not only for maintaining good antioxidants level in the body, but also for dealing with long-term complications that may occur [23]. Finally, a high-fiber diet reduces daily blood glucose levels, post-prandial sugars, and, consequently, HbA1c [79]. To summarize all of the above information, the prepared products contain a unique combination of nutrients that follows the guidelines mentioned above and could be added as new products to the nutritional plan for diabetics.

Except for the nutritional and antioxidant profile, the development of the new products was focused on plant-based sources of nutrients [80]. Increased consumption of whole grain foods, fruits, and nuts significantly helps to control blood sugar. A common element of all of the above foods is their increased fiber content and low glycemic index. The “good” fats in fruits, such as cranberries and apples, appear to be beneficial in diabetes prevention [81]. Vegetarianism also helps patients to better manage the accompanying disorders of diabetes, such as elevated lipids and cardiovascular disease [82], and find alternative sources of protein [83] from either whole food sources or powders containing protein from multiple sources [84,85].

Oat is common in many foods, such as bread, biscuits, and snack bars [86], because of its low GI [87]. A meta-analysis of 103 trials revealed that β-glucan affects blood sugar levels after a meal and found evidence to suggest that carbohydrate-based meals containing β-glucan were linked to lower blood sugar levels than those without it [88]. Also, a review of 16 studies [89] concludes that oats have a beneficial effect on glucose control and lipid profiles in people with type 2 diabetes.

Additionally, the fruits selected for use in the bars have low Glycemic Indexes [90]. A recent study [91] found that patients who ate fresh fruit daily had lower rates of type 2 diabetes. Also, a large study [92] found that people who consumed whole fruits, especially blueberries, grapes, and apples, had significantly lower risks of developing type 2 diabetes. Fruits [70,85] and fruit juices [83], as well as sweeteners, such as honey [86], have also been used in the past by other researchers in the formulation of snack bars, but no-one used carob honey [27], which is the only sweetener specifically used to help patients with diabetes.

Nuts also contain high levels of plant proteins, unsaturated fatty acids, and other nutrients, including vitamins and phytochemicals such as flavonoids, and might be the main reason, besides the presence of *Coprinus comatus* powder, for the oxidative stability of plant-based snack bars (Figure 2). A systemic review [93] concluded that eating nuts either in whole or as unprocessed a state as possible could benefit people with diabetes. Another study has shown that the intake of at least 5 servings of nuts per week (where 1 serving corresponds to 28 g) led to a reduction of up to 27% in the risk of type 2 diabetes [81]. Nuts have also a fatty acid profile that favorably affects blood lipids and lipoproteins. They are low in saturated fat and high in unsaturated fat [94]. The amount of lipids released from the in vitro digestion (Figure 3a,b) and fatty acids was smaller than in all of the commercial bars. Developing a mechanistic understanding of the impact of food structure and composition on human health has increasingly involved simulating digestion in the upper gastrointestinal tract [47]. The structure and formulation of cereal foods affect the rate and extent of starch digestion.

Cinnamon is widely applied in the food flavoring industry. The high acceptance of the aroma characterization by the consumers may stem from the inclusion of this spicy product and essential oils from all of the ingredients. Firstly, Rousel et al. (2009) [95] proved the antioxidant profile of cinnamon and, thus, its main role in diabetes research [96]. The results showed reduced body mass index, especially in people aged 50 years old or under with a body mass index of ≥30 Kg/m^2^ [97]. The dose of the supplementation of cinnamon was ≤1500 mg/day for ≥12 weeks and could reduce body weight. The review by Namazi et al. [98] focused on the role of cinnamon in DM and the results showed a −19.26 mg/dl reduction in the fasting blood glucose. The therapeutic benefit of consuming each of the new bars depends on the concentration of cinnamon, and it was observed from the 112 mg/100 g product.

Finally, the new ingredient—*Coprinus comatus* powder—can characterize an alternative protein. Plant-based proteins present a promising solution to our nutritional needs due to their long history in crop use and cultivation, lower cost of production, and easy access in many parts of the world [74]. The nutritive value of a protein is determined not only based on its amino acid composition, but also its digestibility and absorption as free amino acids. Plant-based proteins of PB snack bars were more resistant to digestion and accessibility to enzyme activity and the same results were observed by other researchers [99]. Also, the antidiabetic properties [100,101,102] could be added to the list of benefits of novel mushroom powder. Moreover, it has been shown in vivo to possess hypoglycemic effects [103]. According to a recent review [31], *Coprinus comatus* contains 14.2 g protein/100 g dry mass, 53.8 g/100 g dry mass carbohydrates (12.3% fibres), and 0.9 g fat/100 g dry mass (75.7% unsaturated fat). Additionally, it is rich in different kinds of vitamins and minerals that are absent in several vegetables and meat. Research suggests that the intake of 100 g/100 g dry mass *Coprinus comatus* contains 30% of vitamin E, C, D and Iron, Selenium, Magnesium, and Zinc, which have a variety of proven antioxidant, anti-inflammatory, and antimicrobial properties [31]. If these nutritional products were added to a person’s diet, they may improve their overall health and protect them from inflammation and many diseases [100,101,102], which, by definition, makes the snack bar a functional food [104]. Therefore, the intake of elements of toxicological importance (Pb, Cd, As) via daily consumption of 30 g portion of dry weight mushrooms poses no risk [105].

Any food that meets any of the following three definitions is called a “novel food”: (a) a food or product that does not have history of safe use as a food, (b) a food that results from a process that has not been previously applied to food, and (c) a food that has been modified via genetic manipulation. Functional foods are novel foods that have been formulated to ensure that they contain substances or live micro-organisms that have possible health-enhancing or disease-preventing value at a concentration that is both safe and sufficiently high to achieve the intended benefit [104]. Thus, according to the above considerations, and taking into account what has been described above, we can state that plant-based (PB) snack bars enriched with *Coprinus comatus* powder are novel foods because they allow us to redesign our food production system by using new sources with low impact.

Developed plant-based snack bars are novel foods due to all of their ingredients (pea or rice protein, agave syrup, carob honey, oat bran, oat flakes, lemon juice and zest, orange juice and zest, almonds, flaxseed, sunflower oil, cranberries, apples, cinnamon, and chocolate with stevia) and the addition of *Coprinus comatus* powder [106], which is not reported in any other snack bar. A protein powder is a dietary supplement. The FDA leaves it up to manufacturers to evaluate the safety and labeling of products. The Recommended Dietary Allowance for protein intake is 46 g per day for women and 56 g for men [107]. It has been reported that edible mushroom extract containing a high amount of L-Ergothioneine, which is received in a concentration up to 30 mg/day for adults and 20 mg per day for children, has been proven not to be genotoxic [108]. Except for nutritional profiling, there were also significant differences between the novel bars and the commercial bars in appearance, aroma, and taste, and the commercial bars received the lowest score for all attributes. Additionally, 65% of consumers seem to prefer plant-based foods. Furthermore, the diabetic food market has grown in size [109], and these products could be accepted due to consumers’ improved perception of them (Figure 2).

Finally, this study focused on the use of low glycemic natural sweeteners in the formulation of two plant-based snack bars and protein source from *Coprinus comatus* as a potential fortifier to enhance the functionality of the snack bars. The formulated plant-based snack bars, especially those that contain rice protein, which was enriched with *Coprinus comatus* powder, can be consumed as a ready-to-eat healthy appetizer for breakfast or in the evening for ensure better glucose control, as they have a proximate composition value and antioxidant profile comparable to that of the commercial bars. Moreover, plant-based products can promote health benefits, including antioxidant properties, as well as glucose control [64,65].

From a physiological point of view, this nutritional intervention can protect against oxidative stress. Patients using intensive insulin therapy regimens who maintain very good control may benefit most from using diabetic snack bars that contain mushroom protein. Individuals who suffer from hypoglycemic unawareness may benefit from the use of snack bars as part of a healthy diet to help prevent episodes of hypoglycemia throughout the day. However, this impact will be evaluated in the future using clinical trials.

The results of the clinical research attempted to highlight the decisive importance of nutrition in terms of determining the course of the disease and the need for patients to adopt correct eating habits. The recording of changes in anthropometric characteristics, as well as biochemical indicators, may reveal the need for specific functional foods and/or dietary guidelines for patients, as well as their possible inability to comply with the expected health consequences, while analyzes are already being performed regarding the micronutrient content in the new products. Ultimately, however, determining overall dietary habits and patterns of food intake over time are critical to nutritional health and success in achieving good glycemic control, rather than the intake of a single food product [18,110,111].

## 5. Conclusions

Plant-based bars were designed and developed, and the end products were studied in the laboratory to determine their properties in regulating sugar, which is a factor that contributes to the management of body weight. These bars were prepared using the traditional mushroom *Coprinus comatus* of Thessaly as a powder, which received a high acceptance by consumers and have great potential to be commercialized as a novel food product. After assessing their nutritional value, they were assigned the following nutrition claim: a “Source of fiber and Source of Protein”. Based on the experimental procedure that simulated the digestion of the formulated plant-based snack-bar with rice protein, which was enriched with *Coprinus comatus* powder, it seems to be slowly absorbed by the body and, unlike other snacks, causes less sugar rise, as evidenced by the low concentration of reducing sugars determined during simulated digestion. Further studies and clinical trials are required to explore the product’s acceptability and determine the health effects of these products in patients with diabetes.

## Figures and Tables

**Figure 1 foods-12-02702-f001:**
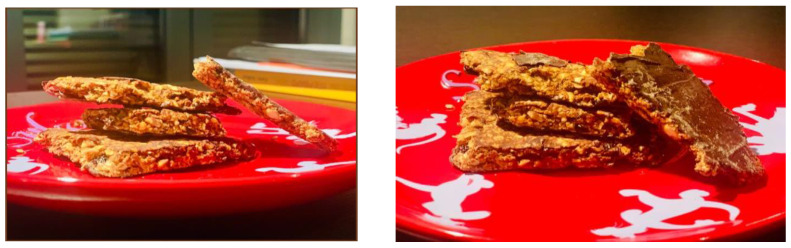
Photograph of baked plant-based snack bars (rice protein and pea protein with *Coprinus comatus* powder respectively).

**Figure 2 foods-12-02702-f002:**
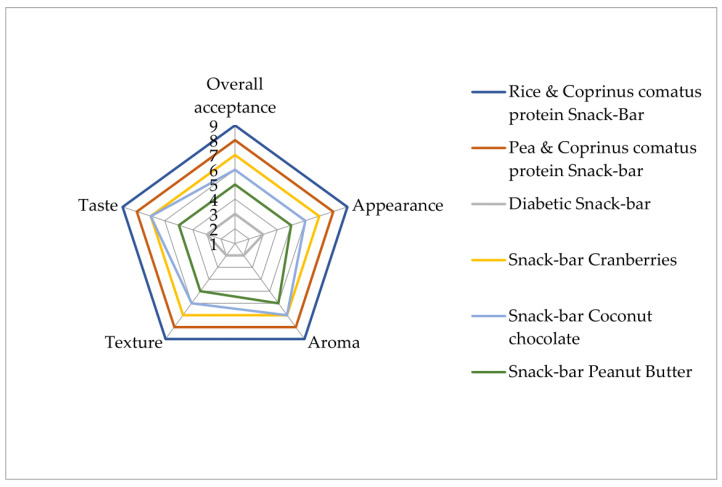
Sensory mean scores (*n* = 20) of experimental plant-based bars and four commercial bars.

**Figure 3 foods-12-02702-f003:**
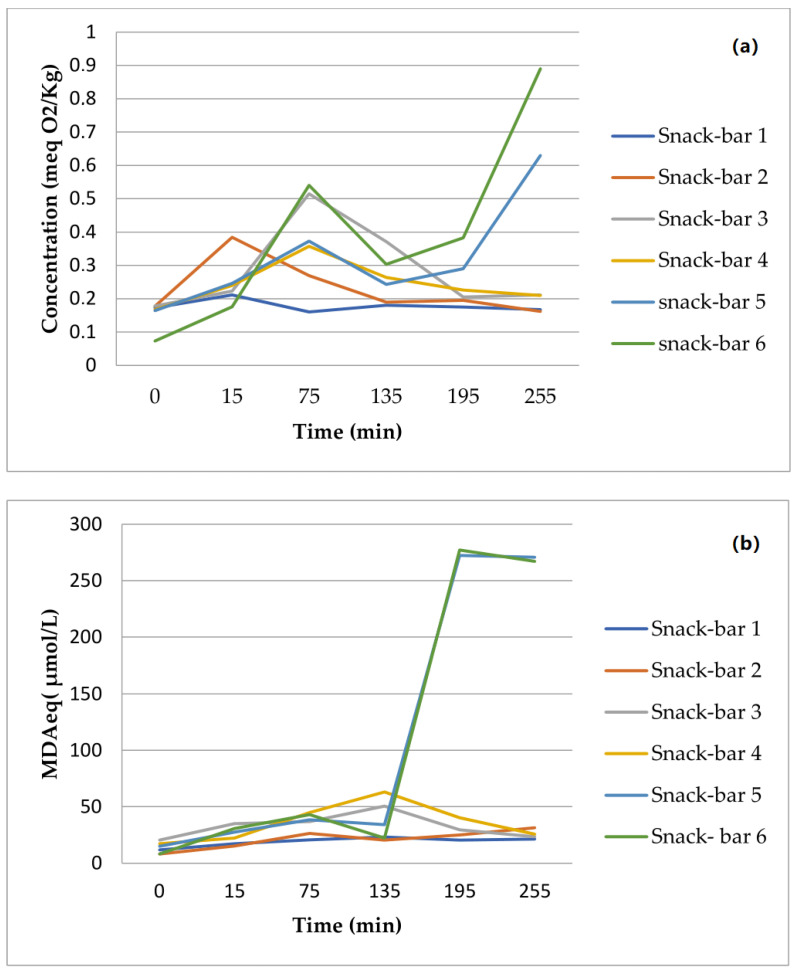
(**a**,**b**) Effect of the GI tract on the oxidative stability of samples (peroxide value indicates primary oxidation products and TBARS secondary oxidation products). Different geometrical forms (one-way ANOVA) denote significant differences (*p* < 0.05) among the six stages for the same compound.

**Table 1 foods-12-02702-t001:** Nutritional value of snack bars (per 100 g product).

Snack Bar to Prevent Hypoglycemia or Reduce Post-Prandial Hyperglycemia	Energy (kcal)	Protein (g)	Carbohydrates (g)	Fibers (g)	Fat (g)
Extend Bar * (Clinical Products, Ltd.) [21]	160	2.5	30	0	2.5
Nite Bite Timed-Release Glucose Bar (ICN Pharmaceuticals, Inc., Orangeburg, NY, USA) [21]	100	3	15	0	3.5
Gluc-O-Bar (Clinical Products, Ltd.) (APIC, Inc., Arlington, VA, USA) [21]	130	7	21	0	2.5
Ensure Glucerna [21]	140	6	24	4	4
Choice DM (Peanut flavor) Mead Johnson Nutritionals [21]	140	6	19	3	4.5
Choice DM Crispy Bars (Ross Products Division) [21]	120	4	21	1	2.5
Nothing-Else [22]	143.3	4.5	17.9	3.3	6.8
Nothing Else-Snack (Re)formulation [23]	131.6	6	23.3	4.3	8.9
Nothing Else-Snack (Re)formulation (ΝΕ) [20]	253.2	11.6	44.9	8.2	17.1

* Snack Bars have the firm of their company, were investigated from MHRA (Medicines and Healthcare Product Regulatory Agency), produced and labelled to United States.

**Table 2 foods-12-02702-t002:** Formulation of PB snack bars.

Ingredients	Rice-Protein Snack Bar	Pea-Protein Snack Bar
Plant-based protein (rice or pea protein respectively) (g)	31.08	31.08
Agave syrup (g)	21.32	21.32
Carbon honey (g)	9	9
Oat bran (g)	46.79	46.79
Oat flakes (g)	43.87	43.87
Lemon juice (g)	2.5	2.5
Lemon zest (g)	1.22	1.22
Orange Juice (g)	10.05	10.05
Orange zest (g)	1.22	1.22
*Comprinus comatus* powder (g)	1	1
Almonds (g)	26.04	26.04
Flaxseed (g)	3.54	3.54
Sunflower oil (g)	30	30
Cranberries (g)	21.23	21.23
Apples (g)	10.7	10.7
Cinnamon (g)	0.4	0.4
Chocolate with stevia (g)	40	40

**Table 3 foods-12-02702-t003:** Proximate analysis of two plant-based snack bars (University of Thessaly) and nutritional value of commercial snack bars (per 100 g product).

Snack Bar ^1^	Energy (kcal)	Protein (g)	Carbohydrates (g)	Fibres (g)	Fat (g)	Saturated (g)
1	424	18.8	39.6	4	20.3	16.2
2	435	19.5	38.4	3.5	22.2	18.3
3	347	4.3	74	4.5	6.8	2.1
4	415	9.6	57.8	8.6	16.2	3.8
5	447	8.6	60.6	11.1	18.4	10.3
6	438	10	59.5	9.7	18.4	4.7

^1^ All bars contain between 60 and 110 mg Na and between 50 and 105 mg K per serving.

**Table 4 foods-12-02702-t004:** Texture profile analysis (avg ± standard deviation).

Snack Bar	Hardness 1 (g)	Hardness 2 (g)	Springiness (mm)	Cohesiveness (g)	Adhesiveness (g)	Chewiness	Gumminess
1	4287.3 ± 122.8 ^a^	2384.9 ± 594.2 ^a^	4.5 ± 1.09 ^a^	0.27 ± 0.07 ^b^	0.8 ± 0.35 ^a^	5075.98 ± 1582.06 ^a^	1128.89 ± 261.03 ^a^
2	3035.6 ± 680 ^a^	576 ± 121.6 ^a^	4.18 ± 0.54 ^a^	0.1 ± 0.02 ^b^	0.41 ± 0.09 ^a^	1173.22 ± 108.42 ^a^	282.71 ± 18.18 ^a^
3 ^1^	-	-	-	-	-	-	-
4	1317.5 ± 111.5 ^a^	798.9 ± 65.4 ^a^	1.88 ± 0.44 ^a^	0.11 ± 0.03 ^b^	0.1 ± 0.04 ^a^	254.95 ± 71.69 ^a^	136.78 ± 30.51 ^a^
5	992.8 ± 23.0 ^a^	679.6 ± 9.5 ^a^	3.30 ± 0.60 ^a^	0.1 ± 0.0 ^b^	0.7 ± 0.4 ^a^	491.1 ± 92.6 ^a^	149.1 ± 13.3 ^a^
6	495.3 ± 24.0 ^a^	371.5 ± 17.8 ^a^	3.07 ± 0.31 ^a^	0.24 ± 0.03 ^b^	1.2 ± 0.3 ^a^	360.93 ± 70.15 ^a^	116.9 ± 12.03 ^a^

Table values are means ± standard deviations. Different superscript letters in the same row represent statistical differences (*p* ≤ 0.05). ^1^ Outside of the range of the Texture Analyzer.

**Table 5 foods-12-02702-t005:** Reducing sugars’ concentrations (g/g bar) during simulated digestion in three phases (oral, gastric, and intestinal phases). Data are expressed as mean values (*n* = 3).

Snack Bar	BOD t = 0 min	AOD t = 15 min	GP t = 75 min	AGP t = 135 min	IP t = 195 min	AID t = 255 min
1	0.166 ^a^	0.115 ^a^	0.033 ^a^	0.064 ^a^	0.056 ^a^	0.037 ^a^
2	0.143 ^a^	0.112 ^a^	0.133 ^b^	0.086 ^a^	0.165 ^b^	0.204 ^b^
3	0.123 ^a^	0.296 ^b^	0.089 ^c^	0.237 ^b^	0.601 ^c^	0.489 ^c^
4	0.156 ^a^	0.292 ^b^	0.111 ^b^	0.213 ^b^	0.205 ^b^	0.358 ^c^
5	0.358 ^b^	0.243 ^b^	0.356 ^d^	0.350 ^c^	0.378 ^d^	0.451 ^c^
6	0.440 ^b^	0.230 ^b^	0.340 ^d^	0.320 ^c^	0.500 ^c^	0.510 ^d^

Different letters (one-way ANOVA) denote significant differences (*p* < 0.05) among the six stages for the same compound. *p*-value < 0.001. BOD, Before Oral Digestion; AOD, After Oral Digestion; GP, Gastric Phase; AGD, After Gastric Digestion; IP, Intestinal phase; AID, After Intestinal Digestion.

**Table 6 foods-12-02702-t006:** Protein content (g/g bar) during simulated digestion in three phases (oral, gastric, and intestinal phases). Data are expressed as mean values (*n* = 3).

Snack Bar	BOD t = 0 min	AOD t = 15 min	GP t = 75 min	AGD t = 135 min	IP t = 195 min	AID t = 255 min
1	0.190 ^a^	0.100	0.128	0.108 ^a^	0.0280 ^a^	0.033
2	0.149 ^a^	0.065	0.084	0.083 ^a^	0.0663 ^b^	0.077
3	0.031 ^b^	0.026	0.0354	0.033 ^b^	0.0112 ^c^	0.009
4	0.078 ^c^	0.029	0.0271	0.039 ^b^	0.0328 ^a^	0.032
5	0.027 ^b^	0.022	0.015	0.011 ^c^	0.0280 ^a^	0.004
6	0.031 ^b^	0.026	0.014	0.096 ^a^	0.075 ^b^	0.013

Different letters (one-way ANOVA) denote significant differences (*p* < 0.05) among the six stages for the same compound. *p*-value < 0.001. BOD, Before Oral Digestion; AOD, After Oral Digestion; GP, Gastric Phase; AGD, After Gastric Digestion; IP, Intestinal phase; AID, After Intestinal Digestion.

## Data Availability

The authors confirm that the data supporting the findings of this study are available within the article.

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
