# Peer review of "Development of a Functional Acceptable Diabetic and Plant-Based Snack Bar Using Mushroom (Coprinus comatus) Powder"

_foods, 2023, doi:10.3390/foods12142702_

Round 1

Reviewer 1 Report

The manuscript has investigated the possible development of functional diabetic and plant-based snack bars using Mushroom (Coprinus comatus) powder. The topic is interesting; However, The manuscript is poorly designed and discussed.

1. L 18, Please express the addition level of mushroom powder in the abstract.

2. What is the best formulation for the preparation of snack bars?

3. Please provide some information about the possible anti-diabetic effects of plant proteins, especially the proteins from rice and pea, as well as mushroom powder.

4. The Materials and Methods section is poorly arranged.

5. Part 2.7; What are the exact percentages of rice, pea, and mushroom proteins in snack bars 1 and 2?

6. L 289-291, For example?

7. L 605-608; Is this phenomenon happening due to the presence of mushroom powder in the rice protein-rich snack bars? Why mushroom does not affect the pea protein-rich snack bars in the same way?

Author Response

The manuscript has investigated the possible development of functional diabetic and plant-based snack bars using Mushroom (Coprinus comatus) powder. The topic is interesting; However, The manuscript is poorly designed and discussed.

  1. L 18, Please express the addition level of mushroom powder in the abstract

We expressed the addition level of mushroom powder in the abstract

  1. What is the best formulation for the preparation of snack bars?

Thank you for the comment we added Table 2 the best formulation.

  1. Please provide some information about the possible anti-diabetic effects of plant proteins, especially the proteins from rice and pea, as well as mushroom powder.

We added the above : Overall, the results indicate that replacing sources of animal with plant protein leads to modest improvements in glycemic control in individuals with diabetes [62]. Plant based protein benefits diabetic people as it is rich in fibre and low in saturated fats. This characteristic of Plant-based proteins helps to reduce the body’s resistance to insulin [63]. Rice and Pea proteins are promising substitutes because of their "allergen-friendly" as well as their emergence in the food market. However, manufacturers generally provide limited functionality information on these proteins [64].

  1. The Materials and Methods section is poorly arranged.

Materials and Methods were re-organized into sections describing: formulation of bars and compositional analysis – attributes – simulated digestion – oxidation determination – and statistics

  1. Part 2.7; What are the exact percentages of rice, pea, and mushroom proteins in snack bars 1 and 2?

Thank you very much for the comment we added the percentages of rice, pea, and mushroom proteins in snack bars 1 and 2.

  1. L 289-291, For example?

We added examples.

  1. L 605-608; Is this phenomenon happening due to the presence of mushroom powder in the rice protein-rich snack bars? Why mushroom does not affect the pea protein-rich snack bars in the same way?

We added the follow explanation: While in a recent study the deconvolution of the amide I band of the Raman spectra indicated pea proteins were different from that of rice proteins and may be that was the reason of differences in our results plus the different functionality of the two PB proteins [64].

Reviewer 2 Report

The manuscript entitled “Development of a functional acceptable diabetic and plant-based snack bar using Mushroom (Coprinus comatus) powder” contains an important study. I have some comments to revise the manuscript.

Abstract:

 Line 27: ‘was observed by’…….please remove ‘was’.

 Line 27: Put a comma after evaluation…..

 Line 28: ‘In conclusion’……please remove this word.

 Line 29-30: Future studies…..I think this should be a part of the conclusions section at the end. Please remove it.

 Introduction

 Line 85: Please correct the extra space after ‘but no proteins’….

 Line 84: In the previous studies……put a comma after studies.

 The functional benefits of mushrooms could be included in the introduction.

 Materials and methods

 Line 119: [31] [32] should be revised as [31, 32]

 Line 120: with stevia [36],….there should be a full stop after [36].

 Line 137: texture profile analysis or TPA should be texture profile analysis (TPA)

 Line 169: by implementing of the………….should be revised as ‘by implementing the’….

 Line 249: ‘International, as’….please remove ‘as’.

 Results and discussions

 Picture 1….Why you have mentioned it as pic 1. Revise it as Figure 1 and make the changes in entire manuscript.

In Figure 2, the statistical errors were not shown. That means the experiments are not repeated thrice.

 Line 422: rucing sugars….???

 Line 440-442: revise the sentence as it is too big.

 Line 444: eliciting greater effects results???

 Line 458:  attributed the great amount of polyphenols…should be revised as ‘attributed to the greater amount of polyphenols’

 Line 460: the flavored…it should be ‘the flavors’

 Line 467: Put a comma after bars.

 Line 475: estimted??? Please correct the other mistakes in this line as well.

 Line 553-556: The sentence seems to be too long. Please revise it.

 Line 565: an functional food???...should be ‘a functional food’

 Conclusions

 Line 599: In the context of this research…….please remove this.

 Line 605-608: The sentence seems to be too long. Please revise it.

  Overall comments:

 Please make the Grammarly check. There are a lot of English errors?

 The critical discussion should be performed.

English needs to be improved significantly.

Author Response

We thank the reviewer for the insightful comments/corrections. Introduction was improved by adding a few comments about the functionality of the mushrooms, as well as about the recently published recommendations of the European Association for the Study of Diabetes (EASD) and the WHO warning for use of non- sweeteners. These, emphasize even more the importance of the newly developed bars, since no artificial sweeteners are used.

A large part of the Results section, it was re-written in an effort to improve the clarity of presentation and avoid grammatical errors.

The manuscript entitled “Development of a functional acceptable diabetic and plant-based snack bar using Mushroom (Coprinus comatus) powder” contains an important study. I have some comments to revise the manuscript.

Abstract:

 Line 27: ‘was observed by’…….please remove ‘was’. It has been corrected.

 Line 27: Put a comma after evaluation….. It has been corrected.        

 Line 28: ‘In conclusion’……please remove this word. It has been corrected.

 Line 29-30: Future studies…..I think this should be a part of the conclusions section at the end. Please remove it. It has been corrected.

 Introduction

 Line 85: Please correct the extra space after ‘but no proteins’…. It has been corrected.

 Line 84: In the previous studies……put a comma after studies. It has been corrected.

 The functional benefits of mushrooms could be included in the introduction.

We added the reference and the main reason we used the certain mushroom.

 Materials and methods

 Line 119: [31] [32] should be revised as [31, 32]

 Line 120: with stevia [36],….there should be a full stop after [36].

 Line 137: texture profile analysis or TPA should be texture profile analysis (TPA)

 Line 169: by implementing of the………….should be revised as ‘by implementing the’….

 Line 249: ‘International, as’….please remove ‘as’.

We chanced all the above!

 Results and discussions

 Picture 1….Why you have mentioned it as pic 1. Revise it as Figure 1 and make the changes in entire manuscript.

In Figure 2, the statistical errors were not shown. That means the experiments are not repeated thrice.

The experiments are repeated thrice. We changed the presentation of the results.

 Line 422: rucing sugars….???  It has been corrected.

 Line 440-442: revise the sentence as it is too big. It has been corrected.

 Line 444: eliciting greater effects results??? It has been corrected.

 Line 458:  attributed the great amount of polyphenols…should be revised as ‘attributed to the greater amount of polyphenols’ It has been corrected.

 Line 460: the flavored…it should be ‘the flavors’.  It has been corrected.

 Line 467: Put a comma after bars. It has been corrected.

 Line 475: estimted??? Please correct the other mistakes in this line as well. . It has been corrected.

 Line 553-556: The sentence seems to be too long. Please revise it. . It has been corrected.

 Line 565: an functional food???...should be ‘a functional food’ . It has been corrected.

 Conclusions

 Line 599: In the context of this research…….please remove this.

We removed it.

 Line 605-608: The sentence seems to be too long. Please revise it.

  Overall comments:

 Please make the Grammarly check. There are a lot of English errors?

  We made all the chances and the check thank you for your help!

The critical discussion should be performed.

We added a paragraph: «Finally, this study focused on the use of low glycemic natural sweeteners in the formulation of two plant-based snack bars and protein source from coprinus comatus as a potential fortifier to enhance the functionality of the snack-bars. The formulated plant-based snack-bars, especially this with rice protein, which was enriched with coprinus comatus powder, can be consumed as a ready-to-eat healthy snack for breakfast or evening for better glucose control as they had a proximate composition value and antioxidant profile comparable to that of the commercial bars. While plant-based products can promote health benefits, including antioxidant properties, as well as, glucose control».

Reviewer 3 Report

In general, the article is interesting and well prepared from the organizational and theoretical side.

My comments

In the discussion chapter, there is a lack of reference of the research results obtained to the results of other researchers obtained in this field. The authors focus mainly on 'promoting' their bar, avoiding scientific polemics.

The question of the origin of the powder from Coprinus comatus needs clarification. Do the authors intend to obtain it in-house. A commercial product (used in the study) may be enriched with substances not directly derived from the mushroom. Therefore, the health results obtained may not be entirely valid.

Author Response

In general, the article is interesting and well prepared from the organizational and theoretical side.

My comments

In the discussion chapter, there is a lack of reference of the research results obtained to the results of other researchers obtained in this field. The authors focus mainly on 'promoting' their bar, avoiding scientific polemics.

The question of the origin of the powder from Coprinus comatus needs clarification. Do the authors intend to obtain it in-house. A commercial product (used in the study) may be enriched with substances not directly derived from the mushroom. Therefore, the health results obtained may not be entirely valid.

Feedback

First of all, thank you for your comments and suggestions.

This is the first study which was used in vitro digestion for snack bars (there were few for black garlic or eggs and some other foods) and that was the reason we didn’t compare the results. We couldn’t compare different foods or methods. Except for the fact that we mentioned “the rice bar shows the best behavior in terms of sugar reduction (Table 6) and is probably more suitable for diabetes’’ and this is an hypothesis according to the analysis of the results from the in vitro-digestion but we would like to make clinical trials in the future as we mentioned at the discussion: «Individuals who suffer from hypoglycemic unawareness may benefit from the use of snack bars as part of a healthy diet to help prevent episodes of hypoglycemia throughout the day. However, this impact will be evaluated in the future using clinical trials». We totally agree with you we also, mentioned in the discussion: «Except for their protein content, cereal bars [18, 22, 23, 54, 71, 72] can be great "anti-diabetic” group of plant foods due to their increased fiber content and their low glycemic index [56]. ».

According to the next comment, powder coprinus comatus produced in a factory according to ISO 22000. ISO 22000 is the only international voluntary standard covering food safety management. It demonstrates an ability not only to identify and control food safety hazards, but also to provide finished, safe products at all times. Moreover, all the ingredients were commercial and had labels as the mushroom powder. CLP (European Union) sets general requirements for labelling to ensure the safe use and supply of hazardous substances and mixtures. Finally, as we mentioned the new plant-based bars were developed in the Laboratory of Technology and Quality Control and Food Safety, of the Department of Agriculture, Plant Production and Rural Environment of the University of Thessaly, according to the rules of Good Hygienic Practice (GHP). The bars for the sensory evaluation were produced on a pilot scale according to ISO 22000.

Thank you for your comments!

Round 2

Reviewer 1 Report

The manuscript is now acceptable.

Author Response

thank you

Reviewer 2 Report

The manuscript can be accepted for publication.

Quality of English Language IS improved.

Author Response

thank you

Reviewer 3 Report

The authors have taken suggestions and comments into account.

I make no further comments.

Author Response

thank you